# Caregiver perceptions and experiences of paediatric emergency department attendance during the COVID-19 pandemic: A mixed-methods study

Gayle Appleby[1‡], Vasiliki Papageorgiou[2‡], Shona Horter[3], Alexandra Wharton-Smith[3], Tina Sajjanhar[1], Anthony Hemeson[1], Emmanuel Singogo[4], Benjamin Cahill[1], Sophie Keers[1], Lorraine Wicksey[1], Marylyn Emedo[1], Alastair Yim[1], Maggie Nyirenda-Nyang'wa[1,5]*

1 Lewisham and Greenwich NHS Trust, London, United Kingdom, 2 School of Public Health, Imperial College London, London, United Kingdom, 3 London School of Hygiene and Tropical Medicine, London, United Kingdom, 4 University of North Carolina Project-Lilongwe, Lilongwe, Malawi, 5 The Infection, Immunity and Inflammation Research and Teaching Department, University College London, London, United Kingdom

‡ GA and VP share co-first authorship on this work.
* mnyirenda@nhs.net

**Data Availability Statement:** Access to the full data set that supports the findings of this study are

## Abstract

### Background

During the early stages of the COVID-19 pandemic, concerns were raised about reduced attendance at hospitals, particularly in paediatric emergency departments, which could result in preventable poorer outcomes and late presentations among children requiring emergency care. We aimed to investigate the impact of COVID-19 on health-seeking behaviour and decision-making processes of caregivers presenting to paediatric emergency services at a National Health Service (NHS) Trust in London.

### Materials and methods

We conducted a mixed-methods study (survey and semi-structured interviews) across two hospital sites between November-December 2020. Data from each study were collected concurrently followed by data comparison.

### Results

Overall, 100 caregivers participated in our study; 80 completed the survey only, two completed the interview only and 18 completed both. Our quantitative study found that almost two-thirds (63%, n = 62) of caregivers attended the department within two days of their child becoming ill. Our qualitative study identified three major themes which were underpinned by concepts of trust, safety and uncertainty and were assessed in relation to the temporal nature of the pandemic and the caregivers' journey to care. We found most caregivers balanced their concerns of COVID-19 and a perceived "overwhelmed" NHS by speaking to trusted sources, predominantly general practitioners (GPs).

available to interested parties on request from the Lewisham and Greenwich Research and Development team (LGT RND) LH.RD@NHS.NET, subject to interested parties having the necessary ethical approval. The full data set is not publicly available in a repository as participant's informed consent did not include an agreement to transcripts being published in full.

**Funding:** MNN received funding from Lewisham and Greenwich NHS Trust R&D Department for the qualitative component of the study. The funders had no role in study design, data collection and analysis, decision to publish, or preparation of the manuscript.

**Competing interests:** The authors have declared that no competing interests exist.

## Conclusion

Caregivers have adapted their health-seeking behaviour throughout the pandemic as new information and guidance have been released. We identified several factors affecting decisions to attend; some existed before the pandemic (e.g., concerns for child's health) whilst others were due to the pandemic (e.g., perceived risks of transmission when accessing healthcare services). We recommend trusted medical professionals, particularly GPs, continue to provide reassurance to caregivers to seek emergency paediatric care when required. Communicating the hospital safety procedures and the importance of early intervention to caregivers could additionally provide reassurance to those concerned about the risks of accessing the hospital environment.

## Introduction

On 11 March 2020, the World Health Organization announced a global pandemic of severe acute respiratory syndrome coronavirus 2 (SARS-CoV-2), a newly emergent coronavirus, which causes a respiratory tract infection known as coronavirus disease (COVID-19) [1]. The COVID-19 pandemic has impacted all facets of life in the UK; this included the introduction of a national lockdown on 23 March 2020 in response to rising infection rates [2]. Subsequently, the UK entered several lockdown cycles in each devolved nation (England, Wales, Scotland, and Northern Ireland), with the introduction of varying levels of government-imposed restrictions. Government messaging focussed on protecting the National Health Service (NHS) from becoming 'overwhelmed' by hospitalisations, evident in the slogan "Stay Home. Protect the NHS. Save Lives" [3]. However, throughout the entirety of the pandemic, hospitals remained open for those requiring emergency medical attention [2, 4].

Despite hospitals remaining open during this first lockdown, there was a decrease in the number of children attending paediatric emergency departments (PEDs) in the UK during this period [5]. This was replicated globally with significant declines in presentation during COVID-19, compared to a year previously [6–10]. For instance, a 1-year retrospective cohort in Greece reported a 59% decline in visits to the PED in March 2020-February 2021 compared to the previous year [10]. A similar pattern was noted during the SARS [11, 12] and Middle East Respiratory Syndrome (MERS) [13] epidemics. However, this differs from paediatric experiences during the H1N1 swine flu outbreak, where children were an at-risk group and an increase in visits to emergency departments was seen [14]. Additionally reports emerged about delays in attendance resulting in poorer outcomes [15]; this was observed for acute paediatric presentations such as appendicitis [16] and severe diabetic ketoacidosis [17]. Further studies investigating delays in hospital presentation during the pandemic have had mixed results. Firstly, the British Paediatric Surveillance Unit survey [18] of paediatricians in emergency departments or paediatric assessment units corroborated findings of delay with 32% (n = 241) observing at least one delayed presentation. Secondly, a multicentre surveillance study found a low prevalence of delay; over 90% of recorded cases (n = 1262) were interpreted as having had no delay in seeking treatment [19]. Thirdly, a study reviewing trends in England during the first peak of the pandemic (January to May 2020) found no excess mortality among 0–15 year olds (case fatality rate <0.5%) [20]. However, these studies have been conducted at different timepoints, using different patient populations across the devolved nations and broadly focus on outcomes at the population or hospital-level.

The decision or action taken for a child to present for emergency care (otherwise known as health-seeking behaviour) is complex. Health-seeking is often driven by "parental triage" including the caregivers' personal anxieties, perceptions of the severity of the illness, assessing available information and ultimately deciding to seek advice from a health professional, such as a General Practitioner (GP) [21–23]. Before the pandemic, most studies exploring health-seeking behaviour in paediatrics were undertaken in the context of high PED attendances [24–28]. One study conducted during the pandemic has reviewed caregivers' views for children admitted to a general paediatric inpatient ward at the end of the first wave of the COVID-19 pandemic in the UK [29]. This study by Watson *et al.* [29] identified that the delay in decision to care-seeking was in part driven by a lack of knowledge but primarily fear. Several factors influenced this fear, such as worries of acquiring COVID-19 or overburdening the hospital [29]. Identified factors that could mitigate against the fear and lack of knowledge included adequate signposting, access to decision-making aids and social support [29].

Therefore, as the pandemic continues there is a need to understand whether similar patterns of behaviour are still being seen in emergency care. This would assist in PED service design and delivery for users (caregivers) whilst transitioning from pandemic response to recovery [30]. Our mixed-methods study aims to determine the impact of COVID-19 on the health-seeking behaviour of caregivers as the pandemic has continued, in relation to accessing and attending PEDs, including perceived barriers and facilitators of health-seeking behaviour.

## Materials and methods

### Study design

This parallel, convergent, single time point mixed-methods study consisted of accessing total hospital attendance figures for set periods in 2019–2020 and analysing data from a survey (**S1 File**) and semi-structured interviews (**S2 File**). These were both designed by the research team. This allowed for corroboration, exploration of complementary and diverging points as well as expansion of ideas [31]. The simultaneous design enables direct contextual comparison of experiences, which is important given the dynamic changeability of the pandemic [32].

### Study setting

The study took place over a 7-week period (November–December 2020) at Lewisham and Greenwich NHS Trust. The Trust comprises two district general hospitals in Southeast London: University Hospital Lewisham (UHL) and Queen Elizabeth Hospital (QEH), Woolwich.

Both hospitals serve a large and diverse population. Lewisham is the fifth most populous borough in inner London [33] serving a population of 303,536; the most common ethnic group is white British followed by black African [34]. In Greenwich, the total population is 286,186 [35] of which just over half, 52.3%, of the population identifying as white British with 19.1% as Black of black African background, 9.8% Asian, 8.5% other white and 4.8% mixed ethnic group [36]. Children and young people (under 18 years) comprise around 22–27% of the population of each borough [33, 37]. When the Income Deprivation Affecting Children Index (IDACI) for each borough is compared to the top 10% nationally deprived areas, Lewisham ranks at number 80 and Greenwich ranks at 129 [38].

### Study population and sampling procedures

Initially, a systematic sample approach was intended to be used to recruit every fifth person contacted to participate in the qualitative aspect of the study, with a view that all caregivers present in the PED would be invited to participate. However, due to the nature of the emergency

environment, it was not possible to invite all who attended in the time period. Therefore, a convenience sampling approach was adopted for participant recruitment. Caregivers were approached and invited to participate by a trained member of the research team who had no direct role in the clinical care delivered to the patient, which was designed to avoid any potential conflict of interest. Caregivers were eligible for recruitment based on whether:

1. They presented to the paediatric emergency department at UHL or paediatric assessment unit at QEH with their child between 9am to 5pm, Monday to Friday

2. Their child was aged under 16 years

3. They did not require a translation service for participation in the study

4. Their child did not receive resuscitation or die in the department

COVID-19 infection control measures were adhered to for all aspects of the recruitment and data collection process. Caregivers could decide whether to decline or participate and complete the survey, interview, or both. Of eighteen caregivers whose responses were recorded as to why they declined to participate in the study the reasons given included: feeling their English wasn't adequate to understand the questions or that they were feeling too worried about their child.

It was anticipated, based on estimated attendance figures, that it would be possible to collect 150–200 survey responses in the time allocated for the study. It was estimated that 20–30 interviews would be required for theoretical saturation (no new themes emerging from the data) [39, 40].

## Data collection

Survey questions included reasons for attendance, illness duration, mode of transport and whether advice had been sought prior to attendance (**S1 File**). A ten-point Likert scale was used to determine worry level when attending the emergency department– 0 (not worried) and 10 (most worried), followed by a quantifier response. The survey was given to the caregiver and collected in the PED or assessment unit (**S1 File**).

Semi-structured interviews were conducted to explore experiences and perceptions of attending during the pandemic (**S2 File**). Caregivers' views were obtained around information received during this period. Input from experts in qualitative design was sought for designing the interview topic guide and to ensure that a standard approach to using the interview guide was adopted by all research team members. A pilot interview was conducted with a caregiver during the training session, the results of which are not included in the study. These approaches helped to achieve face validity of the topic guide and test the acceptability of the questions asked [41]. Interviews were conducted by trained members of the research team, in a private area to maintain confidentiality. If caregivers requested a telephone interview, contact was made within 72 hours of attendance to minimise recall bias. Interviews were audio-recorded and transcribed verbatim by the research team enabling data immersion.

The hospital business analysis team was contacted by the research team for access to internal data (not available in the public domain) relating to total attendance figures, reasons for attendance and other parameters over set time periods in 2019 and 2020 (**Table A in S3 File**). This allowed for comparison between our study data and previous time periods to review trends (**Table A in S3 File**).

## Data management and analysis

All statistical analyses were performed using STATA version 16.0 (College Station, Texas, USA). Frequencies, percentages, median and interquartile range were used to summarise

participants' demographics. We used the Spearman's rank correlation coefficient to compare the degree of worry and illness duration prior to attendance. In addition, we used one-way analysis of variance (ANOVA) to compare differences in mean caregivers' age of those who visited the emergency department by ethnicity (white British compared to other ethnic backgrounds).

For the qualitative data, two researchers (AWS and GA) coded the transcripts using an inductive, emergent thematic analysis approach [42] and created an analytic coding framework. Any disagreements were reviewed by a third researcher (MN). Transcripts were collated and managed using Dedoose (Version 8.3.47) [43]. Three overarching themes emerged with the concepts of uncertainty, disruption, trust, and safety underpinning all three themes:

1.  Lived experience of the pandemic and beliefs/ perceptions around COVID-19

2.  Health-seeking and decision-making process for accessing emergency care during the pandemic

3.  Experiences of hospital attendances during the pandemic

These themes are presented using a narrative approach, with illustrative quotes, as outlined in the results section.

The quantitative and qualitative data was analysed separately with integration achieved at the analysis stage in several ways including for complementarity between the two data sets and initiation whereby discrepancies were identified between the survey and in-depth interview responses [44]. Additionally, as per Sandelowski [45], 'quantitizing' occurred whereby some aspects of the qualitative data (health-seeking behaviour and communication sources) were represented numerically to allow for further interrogation and analysis of the data (**Tables C** and **D in S3 File**). Themes or codes were noted as rows and explored against parental worry levels as identified in the survey. The worry levels were categorised as low (0–3), medium (4–6) and high (7–10). Data are also presented visually in the format of a joint display to aid integration and identify patterns [46].

Data are reported using the Good Reporting of a Mixed Methods Study (GRAMMS) checklist **S4 File** [47].

## Positionality and reflexivity

An understanding of how the research team views the world, their positionality, alongside how this was agreed and negotiated is provided to help inform the reader's understanding of the study [48]. The research team was a mix of clinical doctors, paediatricians and social scientists, which provided a balance of both positivist and constructionist world views. All of the interviews were conducted by the physicians, albeit those not directly involved in the child's care. Nonetheless, this could have impacted the co-construction of the account given by the caregiver through creating a potential power imbalance. Given that the majority of the research team were located at the hospitals where the study took place, this proximity to the participants means an insider view was shared. This emic approach afforded local understanding of the issue but may have hampered the ability for the researcher to ask questions that might have been generated by an 'outsider' [48]. The researchers themselves were living through the pandemic and several researchers felt that with the insights shared by the participants from the study changed their outlook and approach when dealing with cases in their day-to-day practice. In terms of reflexivity when GA and AW were analysing the comments, in addition to using the coding framework, concurrent memos and notes were made of streams of consciousness when links or ideas emerged, then discussed during the analytical process.

## Ethics

Ethical approval was granted by the Cambridgeshire and Hertfordshire Research Ethics Committee, IRAS 287536, ISRCTN12833010. Written informed consent was obtained from all caregivers and assent was obtained from the child when possible. Names were pseudonymised and a further identification number was allocated to those participating in the interviews to maintain confidentiality.

## Results

### Demographic characteristics of participants

A total of 100 caregivers participated in the study; 80 caregivers completed the survey only, two completed the interview only and 18 participated in both.

Caregivers' median age was 37 (IQR: 33–41). The majority (83.3%, n = 80/96) were female. Of the 90% of participants who shared their ethnicity; 68% (62/90) identified as white British or white other, 23% (n = 21/90) Black and 7% (n = 7/90) of other Asian and mixed heritage. For demographic details see **Tables 1** and **2**. The ANOVA test showed no significant mean age differences of caregivers when comparing different ethnicities (white British vs other, p = 0.596).

### Quantitative results

**Attendance patterns.** Combined attendances at both sites fell by 45.9% over a 4-month period of September to December, with fewer children attending in 2020 than 2019 (10,357 in 2020 compared to 19,165 in 2019) over a similar period. In 2019, injuries and respiratory conditions each accounted for around 25% of presentations; whereas, in 2020 injuries as a presentation diagnosis increased to 32% and respiratory reasons decreased to 16% (**Table A in S3 File**).

**Study data.** Our data show that 63% (62/98) attended the paediatric emergency department within 2 days of illness commencement. Arrivals were *via* ambulance (13%, n = 3/98), own cars (57%, n = 56/98) and public transport (13%, n = 13/98). Injuries were the most common clinical presentation type at 30% (n = 29/98), followed by respiratory conditions (11%, n = 11/98).

Two-thirds of caregivers (67.3%, n = 66/98) discussed their child's illness prior to presentation. Contact was most frequently made with the GP (29%, n = 28/98) or NHS 111 (19%, n = 19/98). Six participants attended despite being advised not to.

A total median worry level of 3.5 was reported by 98 participants; reported worries included anxiety about contracting COVID-19 and overusing the paediatric emergency department. However, some caregivers were not worried about attending (22.4%, n = 22/98). There was no correlation between self-rated level of worry about coming to the paediatric emergency department and illness duration ($\rho = 0.15567$).

Caregivers were asked to imagine their actions if the pandemic was not occurring; 32% (n = 31/98) would have presented earlier, 16.3% (n = 16/98) would have preferred to see their GP and for 43% (n = 42/98) there would have been no change.

Survey results are presented in **Table B in S3 File.**

### Qualitative results

We interviewed 20 caregivers between the two hospital sites, **Table 2**. Three participants had current or past work experience in a healthcare setting and six had a friend or relative who currently worked as a healthcare professional. Interviews lasted between 10 and 45 minutes.

We identified three main themes from interviews: (1) lived experiences of the pandemic, including beliefs and perceptions of COVID-19; (2) health-seeking and decision-making during the pandemic; and (3) experiences of hospital attendance during the pandemic. These

**Table 1. Demographic characteristics of all study participants.**

| | Category | Total (%) |
|---|---|---|
| **Total participants** | **Total** | **100 (100)** |
| **Caregiver's gender** | Female | 80 (80) |
| | Male | 16 (16) |
| | Not given | 4 (4) |
| **Caregiver's age** | Under 18 | 1 (1) |
| | 18–24 | 2 (2) |
| | 25–34 | 26 (26) |
| | 35–44 | 48 (48) |
| | 45–54 | 15 (15) |
| | 55–64 | 2 (2) |
| | Not given | 6 (6) |
| **Caregiver's ethnicity** | Asian | 4 (4) |
| | Black | 21 (21) |
| | White British | 35 (35) |
| | White Other | 27 (27) |
| | Other including Mixed | 3 (3) |
| | Not given | 10 (10) |
| **Child's age** | 0–5 years | 53 (53) |
| | 6–10 years | 21 (21) |
| | 11–15 years | 24 (24) |
| | Not given | 2 (2) |
| **Reason for attendance** | Injury | 29 (29) |
| | ENT | 5 (5) |
| | Respiratory | 11 (11) |
| | Cardiac | 0 (0) |
| | Mental health | 3 (3) |
| | GI | 11 (11) |
| | Renal | 3 (3) |
| | Other MSK (not Injury) | 8 (8) |
| | New-born | 6 (6) |
| | Dermatology | 8 (8) |
| | Other/systemic | 8 (8) |
| | Neurology | 3 (3) |
| | Haematology | 2 (2) |
| | Not given | 3 (3) |
| **Mode of transport** | Own | 56 (56) |
| | Public transport | 13 (13) |
| | Ambulance | 13 (13) |
| | Walked | 6 (6) |
| | Taxi | 4 (4) |
| | Lift with family member | 1 (1) |
| | School car | 1 (1) |
| | Not given | 6 (6) |

Abbreviations: ENT, Ear Nose Throat, GI, Gastrointestinal, MSK, Musculoskeletal

themes were underpinned by the concepts of trust, safety (in relation to risk and fear) and uncertainty. The major themes and sub-themes identified are outlined in **Fig 1.**

**Table 2. Summary characteristics of caregivers interviewed.**

| | | Hospital | |
|---|---|---|---|
| Characteristic | Total (%) | UHL (%) | QEH (%) |
| **Total** | 20 (100) | 17 (85) | 3 (15) |
| **Gender** | | | |
| *Female* | 17 (75) | 15 (88.2) | 2 (66.7) |
| *Male* | 3 (15) | 2 (11.8) | 1 (33.3) |
| **Child Age** | | | |
| *0–5 years* | 9 (45) | 7 (41.2) | 2 (66.7) |
| *6–10 years* | 3 (15) | 2 (11.7) | 1 (33.3) |
| *11–15 years* | 7 (35) | 7 (41.2) | |
| *Unknown* | 1 (5) | 1 (5.9) | |
| **Ethnicity** | | | |
| *White British* | 10 (52.6) | 8 (57.1) | 2 (66.7) |
| *British Asian/ Indian* | 1 (5.3) | 1 (7.1) | |
| *White Irish* | 2 (10.5) | 1 (7.1) | 1 (33.3) |
| *White Other* | 1 (5.3) | 1 (7.1) | |
| *Black Caribbean* | 1 (5.3) | 1 (7.1) | |
| *Black* | 1 (5.3) | 1 (7.1) | |
| *Black British* | 1 (5.3) | 1 (7.1) | |

Abbreviations: UHL University Hospital Lewisham, QEH Queen Elizabeth Hospital

The pandemic disrupted the lives of all the participants who were interviewed, and the temporal nature of findings is reflected upon. Temporality can be viewed two-fold; firstly, in relation to the pandemic and virus itself through rapid changes in government guidelines and restrictions, and knowledge. Secondly, temporality can be viewed in relation to the care pathway–from a child developing a worrying symptom or injury, the decision-making process to present to care and finally the experience once accessing emergency paediatric care. Therefore, we present our themes through the caregiver's journey to presenting to healthcare services.

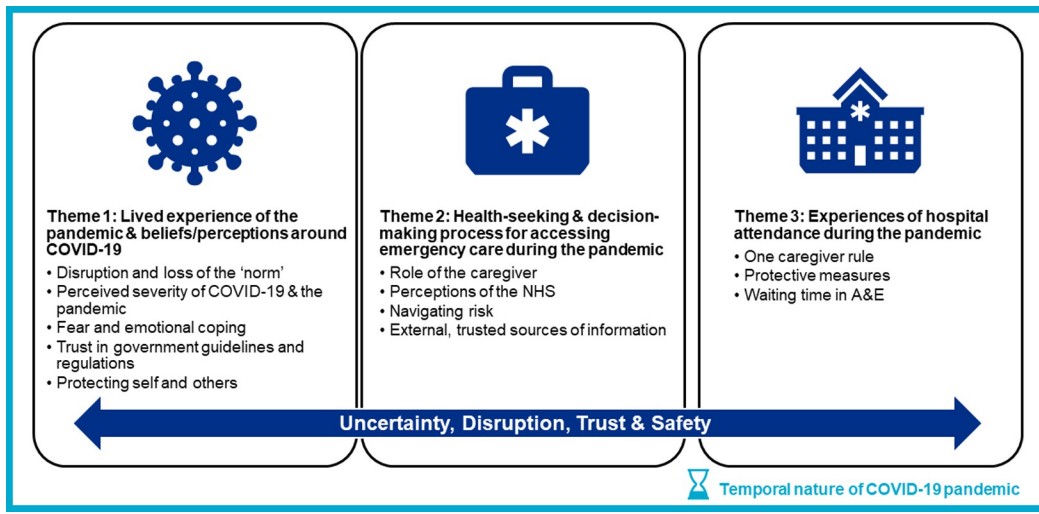

**Fig 1. Conceptual framework including major themes, sub-themes and overarching concepts identified from qualitative data analysis.** Abbreviations: A&E, Accident & Emergency; NHS, National Health Service.

**Theme 1: Lived experience of pandemic and beliefs/perceptions around COVID-19.**
*Disruption and loss of the 'norm'.* COVID-19 presented a disruption to caregivers' lives; for instance, adapting to home working or virtual interactions with family and friends. This disruption left some feeling annoyed and frustrated. Caregivers described a sense of loss being separated from loved ones, loss of 'normalcy' and uncertainty around how long the pandemic may last. However, those interviewed described a level of acceptance of a *"new normal"* and acknowledge that adaptations were necessary to protect public health.

> "*I'm not going to take risks. Even now I. . . try to do most of my shopping online and if I can't get it I'll try to go to the supermarkets where I know the people will be more sensible, i.e. Waitrose or Marks and Spencers (laughs) that sounds like being a real snob but you know knowing that when you go in the door they have the sanitiser and people are generally giving you distance.*" (*ID4*)

> "*Everyone's, been super compliant, poor people having to wear all this stuff [PPE–personal protective equipment] all day long, . . .If they've been taking blood from him—I've been putting . . .my head away from [them. It's] learning how to dance. . .how to dance around it.*" (*ID10*)

*Perceived severity of COVID-19 and the pandemic.* Participants' perceptions of the severity of the pandemic and COVID-19 were often influenced by the 'trustworthiness' of their sources of information. For instance, despite various communication channels such as government, health services, mainstream and/or social media, many participants cited distrust in one or several of these sources. As a result, some described actively avoiding news sources and solely relying on social media due to a perceived vastness of negative news during COVID-19. Some participants referred to their conceptualisation of COVID-19 as a disease and its potential outcomes, such as comparing its severity to flu. When considering this comparison to influenza a spectrum of worry was described with some approaching COVID-19 as they would flu and being unphased (ID12); whereas others were more worried about COVID-19 (ID20).

> "it's just another, another flu or something isn't it" (ID12)

> "*I think it's a case of where people, a lot of people can be not ignorant but are like '. . . it's just a really, really bad cold and it's probably not really any worse than flu' but when people they know start falling [ill]. . . people are scared . . . what if it is COVID?*" (ID20)

Some also spoke of COVID-19 denialism or scepticism in the wider public; for example, believing that some people may be suspicious about COVID-19 if they do not know anyone who has been seriously ill, which could create a level of detachment from the pandemic experience. Additionally, participants alluded to a general COVID-19 'infodemic' of conflicting information where they had to filter misinformation themselves.

*Fear and emotional coping.* Several described feeling nervous, scared, overwhelmed, or exhausted at the time of interview. A notion of 'pandemic fatigue' appeared to exist several months after restrictions were first introduced; however, most described feeling safer (or reassured) over time compared to worried (or fearful) at the beginning of the pandemic. Some participants elaborated on this fear as being linked to the novelty or uncertainty of the pandemic, 'mixed messaging' at the start, concerns for the health of loved ones categorised as 'clinically extremely vulnerable' as well as fears about their own health linked to caregiving responsibilities. For instance, one participant described a heightened phobia of acquiring COVID-19, particularly in public spaces, which could cause stress,

*". . .just generally you feel like everything that you touch is infected (pause), and that is kind of a neurosis of this virus like I feel that everything on the underground is illuminated in it [. . .] it's what it's doing to us mentally, but I wouldn't say just hospitals I would just think anything in a public space. . ." (ID10)*

Another parent described feeling worried about their child being stigmatised for testing positive for COVID-19 within their community,

*". . .because like now if he (son) goes back to school, some parents [. . .] they can say 'oh, y'know so so so has corona, keep away, no'. . ." (ID11)*

Some shared coping mechanisms to deal with their uncertainties and fears which related to faith or religion, hygiene practices and rituals as well as planning to minimise risk.

*Trust in government guidelines and regulations.* Participants referred to rules, to respect the health and safety of others and minimise the risk of onward transmission. Direct references were made to government guidelines, including mandatory face masks in supermarkets and on public transport, social distancing measures and forming 'bubbles'. Specifically, participants described the government guidelines as "confusing" due to their lack of clarity, contradictory changes, or as "unfair" due to 'rule breaks' by members of government, which undermined trust and left participants feeling frustrated (ID8, ID17).

*". . .when we were in the initial lockdown there was less confusion and I think that most people were aware of what was happening and then suddenly we're in another lockdown and everything was rushed and nothing was really broadcast very, very well shall we say and. . . a lot of the time people are unclear as to what they should be doing" (ID5)*

Several parents described the increase in COVID-19 cases among schoolchildren, which was being reported at the time; some specifically expressed their belief that schools should be closed to prevent a surge in cases. One parent explains how he feels confused about how schools appear to be relatively untouched by these rules, which are, in his view, over-implemented in healthcare settings,

*"You can go and mingle at the school gates with fifty or sixty parents in a queue and that's acceptable, but you can't come attend an appointment with your own bubble member. That doesn't make sense." (ID5)*

*Protecting self and others.* Participants recognised COVID-19 as an emerging disease, with knowledge, and information quality perceived as improving over time; however, whether the adoption of preventive behaviours has improved was less agreed upon,

*"I think a lot of people are aware of it, but obviously it all depends if they actually want to listen to the information being given." (ID15)*

Participants described behaviours or practices they had personally adopted to reduce the likelihood of acquiring or transmitting COVID-19, including wearing gloves, staying home, and changing their mode of transport. This varied over time; as time passed, participants felt less worried about acquiring the virus, understood how they could protect others as knowledge about the virus improved, and felt reassured by others adopting preventative behaviours. Participants also presented a narrative of 'social responsibility', agreeing that guidelines and rules

were in place for societal benefit; in other words, to protect others and for the 'greater good' and should therefore be respected. Participant's feelings could also have evolved and adapted as they had been living with the presence of COVID-19 and its impact on their lives for over six months when the interviews were conducted and thereby had had time to adjust to the 'new normal'.

**Theme 2: Health-seeking and decision-making processes for accessing emergency care during the pandemic.**   We investigated how and why caregivers had decided to seek emergency care including influential factors (barriers and facilitators). Participants shared the "trigger" or moment which prompted them to attend the emergency department; often, this related to deterioration of health or pain felt by their child.

Overall, decisions around when and where caregivers sought care for their child were influenced by internal and external factors. Firstly, at an individual level whereby the perceived role of the caregiver to protect their dependent was sometimes referred to as the *"parent's instinct"*, a *"gut feeling/instinct"* (ID14, ID18).

> *". . .so you have that instinct, if you think that your son or daughter . . .is poorly. . . You know . . . then you don't think of anything else apart from getting them the treatment he needs or she needs" (ID12)*

> *". . .I guess you assess as a mum what the circumstances are and if you feel like it's an A&E [problem], I would. . . come to A&E. . . [It's] a gut feeling- when something is serious" (ID18)*

Secondly, balancing risks and contrasting worries such as perceptions of the state of the NHS during the pandemic. Thirdly, the SARS-COV-2 transmission risk (either at hospital or from transport to receiving care) and finally, in relation to information and advice received. Often, feelings of nervousness or uncertainty had resulted in the caregiver seeking an external opinion or validation of the need to seek care and thereby subsequently following this advice.

*Role of the caregiver.* Caregivers identified their responsibility of protecting or negotiating care for their dependents; this was particularly apparent among those who are medically trained themselves or have friends or relatives who work in healthcare services. These 'roles' ultimately fed into their health-seeking behaviour. However, concerns were raised for new parents or those with a limited support networks or living remotely who may not have the confidence to seek emergency care or may be frightened,

> *". . .I've got four children so I feel like maybe I'm not as worried [. . .] I'm also paediatrically first aid trained, I have more confidence in what I would probably see as emergency and not." (ID13)*

*Perceptions of the NHS.* Several parents described the perception of a "stretched" or "overburdened" NHS, either from their own perspective or of other parents, which may influence their decision to present to care,

> *". . .it's [the NHS] overrun anyway, but let alone in a pandemic situation, and you don't want to put additional pressure on the NHS, that was a concern for me before we decided to come, it felt like we needed a medical professional, and I'm. . . glad we did come, but you question yourself, and I'm not qualified to make the decision, but I knew she needed to be seen." (ID7)*

Some perceived social/mainstream media as using scaremongering tactics to generate fear among the public, which was potentially deterring individuals from accessing services. Several participants referred to a hyperawareness of mortality, specifically through media reports of

the large death toll at the beginning of the pandemic, which appeared to influence their decision to present for healthcare,

> *"At the beginning y'know the death rate was high, and nobody wanted to come to the hospital [. . .] but now you can come to A&E even if you're sick, you just have to follow the procedures."* (ID11)

Other caregivers gave specific examples that they had read or heard about; ID16 referred to a perception that some feel they *"don't deserve an appointment"*. News of individuals deterred from seeking healthcare had left ID19 feeling sad and concerned about preventable outcomes,

> *"I thought it was really sad to be honest [. . .] I just couldn't imagine a lot of people doing that, thinking that 'I don't want to be a bother, or it's just going to be bedlam going into a hospital right now.' And you know potentially dying or becoming seriously ill with something that could be treated or prevented entirely you know. . ."* (ID19)

Additionally, participants described a responsibility to "follow the rules" or being "socially responsible"; and avoiding an "overburdened NHS" by managing the condition at home and self-medicating where possible,

> *"So, if I can [. . .] hold the pain, or deal with the sickness, I try and encourage the children to, we try to deal with things at home as much as we can, to try and not have to come to hospital."* (ID13)

*Navigating risk.* Participants also referred to the physical hospital space relating to safety and balancing risks, particularly potential exposure to COVID-19 in areas of the hospital (e.g., waiting area) as influencing their decision to seek healthcare. Mostly, participants described feeling safe due to implemented COVID measures such as social distancing. One parent (ID13) felt reassured by the limited number of people in paediatric A&E. However, whether partners and other children could attend appointments was confusing for some. Additionally, some expressed concerns about people not wearing face masks, who may or may not be exempt, and who may or may not have COVID-19, which affected their feelings of security in the hospital space. Certain participants described not feeling "100% reassured" to attend emergency care but having done so out of necessity (ID20). For one parent (ID5) this was particularly concerning when observed among staff members,

> *"When I come into a waiting room and there are the majority of people not wearing masks and people coughing and I can't control where someone is going to sit or what they are going to do I find that quite distressing particularly bringing a new-born into that environment that I can't control"*

One caregiver also referred to concerns of the unintentional onward transmission of the virus to loved ones which had left them feeling they were balancing the health of their child against that of acquiring the virus from the hospital setting and transmitting this onwards,

> *". . .so definitely feeling like a personal responsibility to not let my nan die by just going to the hospital for whatever reason or you know that's how we felt about it, quite personally responsible.'* (ID16)

This quote echoes comments made by the Health Secretary at the time, MP Matt Hancock, on a radio channel of "don't kill your gran by catching coronavirus and then passing it on" [49].

*External, trusted sources of information.* Reliable sources of advice were identified as referrals from NHS 111, walk-in clinics, pharmacies, educators (where the accident had occurred in a school setting) and medical professionals (e.g., GP), including friends and family members who were NHS workers. Often, speaking to a trusted source acted as reassurance that they were *"not wasting A&E's time"* and that the necessary safety protocols were in place,

> *"I've got a friend who is a nurse [. . .] I did reach out to her and say, 'what if I need to go to A&E?' and she said, 'if you need to go in you go, phone 111 first, ask them what is happening and if you need to go [. . .] That's it there are no ifs, buts, or maybes you've got to try to keep everything going'." (ID9)*

Most spoke of receiving information and updates from their GP *via* text which was considered a trustworthy, helpful, and reassuring source. However, some participants had negative perceptions of accessing care through their GP,

> *"I don't think there is a role for them [GP] anymore because I don't think they, they're serving a purpose. They must be serving some people seeing as they're on the phone all the time" (ID16)*

One interviewee described a sense of "confusion" about deciding the best care pathway; for instance, going to the GP or calling NHS 111, and the disconnect in communication between the two whereby the same conversations may end up being repeated as she was unsure whether they would be "on record". Frustrations relating to NHS 111 were in light of issues with access as well as experiences of call handlers and no follow-up, including from past experiences,

> *". . .no-one has ever been able to get hold of them since this started" (ID16)*

> *"Well, I just think you are calling a service then you should feel that you have been serviced rather than. . . this vast questionnaire and then bumph it just it really didn't end, it didn't really have a conclusion" (ID10)*

**Theme 3: Experiences of hospital attendance during the pandemic.** We asked caregivers about their experiences in the emergency room, including the waiting time. Overall, most caregivers were satisfied with the care they received and spoke of a duty to reassure other parents to attend when needed; however, some conditioned their responses, stating that attending A&E was a "judgement call" that each parent would need to make for themselves based on the severity of their child's condition or after consulting a GP,

> *". . .only if they were severely unwell and if it was something that. . .could[n't] wait for the GP because I feel. . .the trauma. . .associated through this can do more harm than good." (ID5)*

*One caregiver rule.* Several caregivers alluded to a strain that was put on their hospital visit by the "one caregiver rule" that was imposed to promote social distancing by minimising numbers in the waiting area. Several gave an example of how this altered their health-seeking behaviours with one describing how a friend needed to find a babysitter for their other children to enable them to bring in the index case (ID16). Whilst another described how they altered their information seeking behaviour and decided not to Google information that they had been told about their child but reply solely on the medical team's expertise:

*". . .ordinarily I would look at the internet but, in the circumstances, where you have limited support around you. . . I have just chosen to eradicate that sort of self-diagnosis. . .' (ID10)*

*Protective measures.* Caregiver behaviours when attending hospitals mirrored that of their lived experience especially regarding adherence to protective measures they had adopted. One caregiver describes a need to disinfect themselves after attending:

*". . .It's just a case of when I leave, I will be washing and washing and scrubbing myself from head to toe (laughs)" (ID20)*

*Waiting time in A&E.* Several participants referred to their expectations and experiences of waiting times in A&E which was often an extension of worry around the child. A common perception was a willingness to wait "as long as it takes" to be seen and being patient for hospitals running at reduced capacity but also understanding that times were dependent on triage assessments and the severity of other children presenting for care. Some caregivers would wait indefinitely, irrespective of the pandemic,

*". . .if I'd made that initial decision to overcome those initial barriers to come. . .then it would be for a serious enough reason that would warrant staying" (ID5)*

However, others spoke of patients who had left A&E after waiting two to three hours and not being seen. Participants recommended more frequent updates (by the hour) where waiting times remain lengthy, with one parent (ID16) suggesting updates could be given by a nurse.

## Mixed-methods results

Comparing and contrasting the quantitative and qualitative results allows for construction of a fuller picture around health-seeking behaviours of caregivers during the pandemic and can explain a more nuanced view than either dataset alone, **Fig 2**.

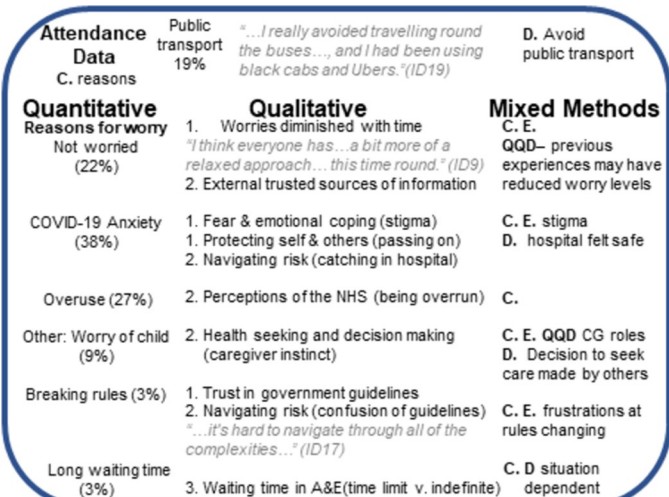

**Fig 2. Mixed method joint display to illustrate how quantitative and qualitative data were integrated.**
Abbreviations: C, Complementary; D, Divergence; E, Expansion; NHS, National Health Service; QQD, Quantitizing Qualitative Data. Three themes: 1. Lived experiences, 2. Health seeking and decision making 3. Experiences of hospital.

Complementarity is seen between attendance reasons of our study data as compared to the 2020 figures. Reduced respiratory attendances noted for both our data and 2020 figure compared to 2019. However, when attendance modalities were compared from the quantitative study to 2019/2020 figures even though public transport and taxi use increased this may be a divergent finding or skewed by increased taxi use. This is because in the qualitative interviews people gave examples of preferring to avoid public transport where possible.

Complementarity is also seen around topics such as the expressed worry of caregivers' and reasons for this including concerns of acquiring COVID-19, as demonstrated in the survey. This view is echoed in the qualitative data and can be extrapolated/expanded as concerns extend not only in the moment of the attendance itself but also prior to attending (e.g., selecting mode of transport and/or worry around onward transmission of COVID-19 following hospital attendance). An additional concern around the stigma of having COVID-19 was also identified (2/20). A divergent view emerged with two (2/20) caregivers noting that whilst they were concerned about COVID-19, and the risk acquisition, they recognised the measures taken to make hospitals safe and felt safer there than other public spaces. This represents a concept of symbiotic safety and trust placed in the hospital setting and those who work there

*". . .it is good to know that hospitals are taking precautions. . . I imagine that the doctors and nurses. . .don't want to catch this so I'm assuming that there are. . . things in place keeping everyone safe." (ID18)*

The safety measures in place may offer one explanation as to why people expressed in the survey they were not worried in attending (22%). Reasons for this can be expanded from the qualitative data and include that caregivers were reassured by the advice they had received from other sources such as GP or 111. An additional perspective is provided when reviewing the quantified qualitative data analysing previous attendance needs during the pandemic. Of the sixteen participants who were asked, a quarter (4/16) had sought acute medical attention earlier in the pandemic. Three of these participants reported low worry scores and their prior attendance experiences may have helped ease their worries when attending during the study:

*". . .I think the third time I came in today I felt probably the least nervous probably because I had already done it." (ID6)*

This contrasts with a caregiver who had high worry level and had previously specifically avoided attending the emergency department:

*". . .we went to a walk in instead of the hospital, because it's quieter, . . . you wait in the car park in the car and then they called us, so we didn't have to wait in the waiting room. . ." (ID13)*

A further example of the complementary aspect of the datasets are around the behaviours of caregivers; the survey describes several different imagined behaviour states if the pandemic had not been occurring, including how 42% of participants said their behaviour would not have changed. This aligns with phrases from the interviews detailing how parents were focussed on acting in the best interests of their child irrespective of the concerns, if any, that they had around the pandemic.

Caregivers in the survey expressed that worry about their child was a concern when attending the hospital (9%). This was also expressed in the interviews with several caregivers talking about following their gut instinct (9/20) due to worries about their child. However, the

quantitized qualitative data reveals a divergent situation emerged whereby it wasn't caregiver choice to seek help but rather it was suggested by another agency, in both cases it was school (**Table C in S3 File**).

> *"So, she got sick at school and they called the ambulance, they've been calling the ambulance for her. So, if she was at home with me, I maybe wouldn't have called 999 or brought her in, I would have just dealt with it myself, which is what I have been doing" (ID13)*

In the survey, 'rule breaking' was identified as a potential worry in attending. In the interviews, complementarity was seen alongside expansion of caregivers expressing frustrations at the ever-changing rules. Long waiting times were also identified as a possible worry (3%). Whilst some caregivers agreed with this sentiment in the interviews a divergent point of view emerged from others where they would be prepared to wait for as long as necessary, which reflects the complexities and multi-factorial nature of health-seeking.

Of the sixteen caregivers who were asked directly in the interview if they recalled receiving information about attending the emergency department in the pandemic, six recalled receiving information and 10 did not. From several of the interviews, a narrative emerged around the perception and/or relevance of the information to themselves and their situation. Several caregivers initially said they had not received information but later commented they had seen and/or heard about information which perhaps suggests at the time of receiving the information it was not relevant to their needs,

> *"Well, I suppose there is what you hear on the television and the media and how you know of overflowing A&Es. . . But I personally have not looked for information." (ID18)*

The most common sources of information mentioned in the interviews were NHS 111 (13/20, 65%) and GP services (9/20, 45%). This corroborates with the overall quantitative findings that these two services were the most common sources that caregivers consulted prior to attending the hospital. However, in addition to being a source of advice, a further divergent view emerged from the qualitative interviews. Specifically, three participants described their belief that they needed to contact their GP or 111 before attending A&E; therefore, these services are perceived to be a gatekeeper to A&E attendance,

> *"I did hesitate and not come straight; I called 111 first. Cus that was what I heard before you come to A&E ring 111" (ID2)*

Similar to the quantitative data, other sources of information were also described in interviews; one-third of participants mentioned other sources including social media (7/20, 35%), television news (7/20, 35%), online resources (7/20, 35%) government briefings (6/20, 30%) and friends and family (7/20, 35%). Resources that exist but were infrequently mentioned were doctors working in hospitals (2/20 10%), pharmacists (1/20, 5%) and government websites (2/20, 10%) (**Table D in S3 File**).

When interview participants were asked to imagine how they would like information to be shared about the pandemic and health seeking, no consensus view was generated. Common responses were social media (7/20, 35%) and mainstream media (8/20, 40%), with only one person feeling that official government resources would be beneficial. When considering the health services and the role they had to play in sharing of information the participants mentioned several community services, which cumulatively make this the largest group consulted. This includes GPs (9/20, 45%), health visitors (4/20, 20%), district nurses (1/20, 5%) and NHS

111 (7/20, 35%). Indeed, this community approach also extended to include public space advertisements including billboards or bus shelters (9/20, 45%). Leaflets were not a popular suggestion with three people actively against this (**Table D in S3 File**).

## Discussion

### Key findings

COVID-19 has disrupted the lives of caregivers and has resulted in feelings of uncertainty about when and how best to seek emergency paediatric care. Our quantitative study indicates a reduction in attendance to the paediatric emergency department during COVID-19. This may be due to fears of attending due to COVID-19 either from personal experience, media portrayal, or concerns of burdening the NHS [29]. Interviews were conducted 8-months into the pandemic (November 2020) and we find that participants had begun to adapt and 'accept' some of the changes to daily life (e.g., government restrictions, public health guidance). However, fears around the uncertainty of the pandemic and seeking healthcare appears to have remained.

Our qualitative study identified several factors as influencing caregivers' health-seeking, among those who accessed paediatric emergency care in London during the "second wave" of the COVID-19 pandemic in the UK. It is important to note that interviews were conducted at a time of increased cases being reported and had necessitated a second lockdown, whereby London was particularly affected, with some hospital sites across the city restricting paediatric emergency services [50]. Caregivers remained confused and fearful in their decision to seek healthcare whilst balancing the risks of acquiring COVID-19 in the hospital setting alongside perceptions of an overburdened NHS. Specifically, three core concepts underpin the experiences of caregivers' lives during the pandemic and decisions to present to emergency care: (1) uncertainty and disruption, (2) trust and (3) safety. Firstly, the uncertain and unpredictable nature of life during a pandemic has inevitably impacted caregivers, who referred to early feelings of confusion due to mixed messaging from the government. This supports findings previously reported by Breckons *et al* [51]. However, it appears that worries and fears about COVID-19 improved as government restrictions eased and caregivers adapted to the 'new normal'.

Secondly, we find that fears of the virus and pandemic, in relation to acquisition and transmission, were outweighed by the emergency nature of the child's condition and caregivers' concerns for their child's health. Our findings align with evidence of the factors driving hospital attendance, including parents' concerns for their child, need for reassurance, the urgency of the condition, perceived higher quality of care in PED, difficulties accessing GPs, and benefits of social networks [24–28]. Thirdly, trust and knowledge appeared to help caregivers to overcome their fears; for instance, some sought reassurance from a trusted source, including medical professionals (e.g., GP, friends or family members who were NHS workers), which aligns with the 2020 Ipsos Mori Veracity Index findings [52] or official channels (e.g. NHS 111). Reassurances are particularly related to the safety of the hospital environment. Some caregivers reflected on 'trigger' moments which had prompted their decision to present to care, largely relating to the child, such as their condition deteriorating or experiencing pain. Finally, once attending emergency care, caregivers were generally willing to wait to be seen.

Previous studies examining behaviours during the COVID-19 pandemic have reached differing conclusions when using the Three Delays Model of Health seeking [29, 53]. This model identifies three areas of possible delay: (1) deciding to seek care, (2) reaching the health care facility and (3) receiving care [54]. One study concluded that caregiver behaviours correlated to all three aspects of the model [29], whilst another only identified delays around patients deciding to seek care [53].

Applying our quantitative study findings to this model, a mixed picture emerges. For instance, 31% of respondents reported they would have attended earlier if the pandemic had not been occurring. However, 43% of our population reported that the pandemic made no difference to their attendance; 62% of children were bought within two days of their illness starting and some attended despite being advised not to, which on balance suggests that their health-seeking was not delayed. When applying findings from our qualitative arm to the model, we found that the largest point of delay tended to be around the decision to seek care. At this point, caregivers have to balance a number of internal and external conflicts; several factors described match those outlined by Watson *et al* such as media reports, fears of acquisition and external information sources [29].

## Strengths and limitations

Our study provides unique insight into experiences within the emergency department during the second lockdown of the COVID-19 pandemic in the UK. A qualitative inquiry has enabled us to explore the "how" and "why" behind changes in behaviours, social responses and social interactions during the pandemic [55, 56]. The approach has also allowed us to explore caregivers' health-seeking and decision-making processes which will enable services and approaches to be adapted to better meet their needs. Most interviews were conducted face-to-face which allowed for the interviewer to respond to the body language of participants, although face masks may have interrupted the ability to respond to changes in facial expression [57].

We identified limitations relating to participants' recruitment and data collection. The survey design (**S1 File**) could have been improved by piloting the survey alongside involving a quantitative expert at the design phase [58]. The involvement of an expert coupled with administering a pilot of the survey before data collection started and gaining feedback on acceptability would have allowed face validity to be achieved [41, 58, 59]. Study participants were selected using convenience sampling and it is acknowledged that bias-removal benefits are lost when compared to the initial proposed systematic sampling method [59]. However, when the results are reviewed a spectrum of responses was achieved and not just limited to extreme views. In addition, in a sometimes busy and fast-paced health care setting where speed of seeing patients is prioritised inviting all who attended was not possible and by extension, a simple recruitment strategy was required. In addition, the responses are also limited to those who attended for paediatric emergency care; therefore, the perspectives of caregivers who perhaps should have attended but did not, are absent. Participants were interviewed by clinicians, albeit those not directly involved in the child's immediate care in the acute situation, and therefore the findings may reflect a narrative that is perceived as socially desirable and more acceptable to health providers [60, 61]. This may be particularly the case among those who indicate COVID-19 denialism, with alternative narratives and criticism potentially more difficult to access with this approach. Our findings are relevant to a local population (of Lewisham and Greenwich) but may not be generalisable to other areas. A further limitation was that data collection was limited to weekdays and in hours, therefore those who accessed the emergency department outside of these times were not included. In part, this reflects the historical approach of how research is conducted at the trust as most research studies occur during 'office hours'. Certainly, pre-pandemic studies around PED attendances found evenings and weekends to be busier for paediatric attendances [21]. Therefore, the generalisability of our study is again limited as it does not include those caregivers and children who attend 'out of hours'. Indeed, they may well have had differing experiences of the waiting room and/or expressed additional factors determining attendance including the ability to attend because

their job ended, or school had closed. Finally, it is important to note the context in which the interviews were conducted; often between clinical investigations in the department under time constraints which created challenges for holding longer, more in-depth interviews.

## Future research and study implications

Future studies could explore how the socioeconomic profile of caregivers influences health-seeking decisions during the pandemic. For instance, one study has reported how some sub-groups of caregivers (e.g. mothers, caregivers with children with chronic illnesses and caregivers who are worried about missing work) may be more or less likely to delay presenting to emergency departments due to fears of acquiring COVID-19 [62]. Therefore, communicating risks such as delaying presentations during an emergency, honestly and effectively through targeted and tailored messaging to specific subgroups of caregivers may help alleviate specific concerns.

Besides targeting information to specific population groups another potential implication for future policies and planning that became apparent from the study was around how information is best communicated. Several aspects emerged that were important regarding information delivery including: content, who delivers it and the delivery modality itself. These factors all combine to influence and affect how the information is not only received but subsequently acted upon. Our research shows that whilst a plethora of health-seeking information sources are available to caregivers they valued when information was provided by health professionals and their own social networks with an emphasis on community services as trusted sources. This aligns with the literature on health promotion and the importance of engaging community leaders and key stakeholders, among other recommendations [63]. However, other sources, including government and social media, were met with varying degrees of belief and scepticism. Other authors have also encountered how multiple information sources can cause misconceptions to spread and have argued that it is the contradiction in messages that are delivered that determines behaviours [64]. Certainly, the interviews did reveal confusion and frustrations around the clarity of messaging and being honest in communications. As outlined in the literature, being transparent and rapidly moderating misinformation is another key area to address to take forward in similar situations [63]. Equally, in future pandemics or health emergencies, medical professionals should anticipate being called upon in both a professional and a personal capacity to provide information as it is known and needed. However, there was not a one size fits all approach as to how to best achieve this from our research.

## Conclusion

The uncertainty and disruption of COVID-19 reportedly influenced the health-seeking behaviour of caregivers accessing paediatric emergency care at two hospitals in London. Fear of acquiring COVID-19 in the hospital space and conflicting messaging during the early phases of the pandemic continue to influence decision-making processes. However, if caregivers have any concerns in attending, these are generally supplanted either when their child's health deteriorates and/or once they receive validation from an external, trusted source or individual. Expert opinions of health professionals are prioritised by caregivers, particularly if they have a personal relationship with them (e.g., a friend or relative). However, caregivers continue to share a narrative of wanting to protect an "overburdened NHS", which has dominated government messaging. Therefore, we recommend GPs, and other medical professionals, utilise their trusted position to continue to encourage caregivers to seek emergency paediatric care when required. They should highlight that hospitals remain safe spaces and that the NHS will prioritise their child's care whilst the COVID-19 pandemic continues and beyond.

## Supporting information

**S1 File. PPEDiC survey.**
(DOCX)

**S2 File. PPEDiC interview topic guide.**
(DOCX)

**S3 File. Supplementary data.** Table A Attendance figures compared to study data. Table B Responses to questionnaire compared between the two groups. Table C Quantitizing of the qualitative responses. Table D Quantitizing of qualitative responses around sources of information.
(DOCX)

**S4 File. GRAMMS checklist.**
(DOCX)

## Acknowledgments

We would like to thank all the caregivers who participated in the study. We would like to thank Lewisham and Greenwich Research and Development team for supporting the study. In addition, we would like to thank the hospital business analysis team for their help with sourcing the attendance data. We would like to thank Mary Sawtell and Meena Khatwa for their comments and suggestions during the initial study protocol development period.

## Author Contributions

**Conceptualization:** Tina Sajjanhar, Anthony Hemeson, Benjamin Cahill, Sophie Keers, Lorraine Wicksey, Maggie Nyirenda-Nyang'wa.

**Formal analysis:** Gayle Appleby, Shona Horter, Alexandra Wharton-Smith, Emmanuel Singogo, Lorraine Wicksey, Marylyn Emedo, Maggie Nyirenda-Nyang'wa.

**Investigation:** Gayle Appleby, Tina Sajjanhar, Anthony Hemeson, Benjamin Cahill, Sophie Keers, Lorraine Wicksey, Marylyn Emedo, Alastair Yim, Maggie Nyirenda-Nyang'wa.

**Methodology:** Shona Horter.

**Writing – original draft:** Gayle Appleby, Vasiliki Papageorgiou, Shona Horter, Alexandra Wharton-Smith, Lorraine Wicksey, Marylyn Emedo, Maggie Nyirenda-Nyang'wa.

**Writing – review & editing:** Gayle Appleby, Vasiliki Papageorgiou, Shona Horter, Alexandra Wharton-Smith, Emmanuel Singogo, Marylyn Emedo, Maggie Nyirenda-Nyang'wa.

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
