## [Decision Letter · Decision Letter 0]

23 Feb 2022

PONE-D-21-33341

Caregiver perceptions and experiences of Paediatric Emergency Department attendance during the COVID-19 Pandemic: a mixed-methods study

PLOS ONE

Dear Dr. Maggie,

Thank you for submitting your manuscript to PLOS ONE. After careful consideration, we feel that it has merit but does not fully meet PLOS ONE’s publication criteria as it currently stands. Therefore, we invite you to submit a revised version of the manuscript that addresses the points raised during the review process.

Please read and respond to all of the peer review comments. You should provide a point-by-point response to explain any changes you have (or have not) made to the original article and be as specific as possible in your responses. Before your re-submission, please go through the submission guideline and make sure that the format of the revised manuscript is in accordance with the journal requirement.

We look forward to receiving your revised manuscript.

Kind regards,

Alison Wang

Academic Editor

PLOS ONE

https://journals.plos.org/plosone/s/fileid=ba62/PLOSOne_formatting_sample_title_authors_affiliations.pdf".

2. We noted in your submission details that a portion of your manuscript may have been presented or published elsewhere. [Initial findings of quantitative and qualitative work presented at the RCPCH online conference 2021 and published abstracts in Archives of Disease in Childhood. Both have been uploaded as "Related Manuscript" file types. These are not dual publications as these were top-line initial findings presented for each element of the study; here, our paper presents a complete overview of both the quantitative and qualitative studies.] Please clarify whether this [conference proceeding or publication] was peer-reviewed and formally published. If this work was previously peer-reviewed and published, in the cover letter please provide the reason that this work does not constitute dual publication and should be included in the current manuscript.

**Additional Editor Comments**:

1. Abstract: please remove the "discussion" section from the Abstract

2.This is a mixed method study design, however, it reads like a multiple method study as there is no integration of the quantitative and the qualitative study findings (quantitative and qualitative results were analysed and reported separately in this manuscript). Integration is the key difference between mixed-method and multiple method study design, so please carefully address this issue in your revised manuscript.

**Reviewers' comments:**

**Comments to the Author**

1. Is the manuscript technically sound, and do the data support the conclusions?

Reviewer #2: Partly

Reviewer #3: Partly

Reviewer #4: Yes

2. Has the statistical analysis been performed appropriately and rigorously? 

Reviewer #2: Yes

Reviewer #3: N/A

Reviewer #4: Yes

3. Have the authors made all data underlying the findings in their manuscript fully available?

Reviewer #2: No

Reviewer #3: Yes

Reviewer #4: Yes

4. Is the manuscript presented in an intelligible fashion and written in standard English?

Reviewer #2: Yes

Reviewer #3: Yes

Reviewer #4: Yes

5. Review Comments to the Author

**Reviewer #1**: Thank you for the opportunity to review this manuscript. The paper outlines a mixed-methods study which sought to examine the impact of health-seeking behaviour of parents accessing care in the paediatric emergency department during the pandemic. Beyond a few key concerns, which I will outline below, I think the study is well designed and conducted, however, it fails to outline how the research builds on existing evidence on this topic and whether there are any implications for research, policy, or practice going forward.

Introduction

When identifying gaps in the literature, the authors state “as the pandemic continues there is a need to understand whether the same patterns of behaviour are being seen in emergency care, particularly whether young children are experiencing unintended consequences due to delayed presentation”. However, it is unclear how the present design and results identifies unintended consequences so I would suggest re-wording this as it may be misleading for the reader.

Methods

Study Design and study population:

The authors state that the mixed methods approach was selected because it “allowed for corroboration, triangulation, exploration of complementary and diverging points as well as expansion of ideas”. However, it is not clear from the results how this was carried out as it appears that the quantitative and qualitative data were analysed separately. It could be that I’m interpreting this incorrectly but in that case, this section needs to be re-worded.

Were all attendees to the ED during the timeframe described asked to participate? If not, how were they selected and was there a risk of any bias? What was the response rate?

Data Collection:

More information of the design of the survey is needed. Was it piloted ahead of the study? Any information on validity?

The authors mention that triage scores and outcomes were noted. Were these extracted from medical notes? If so, this needs to be stated as a source of data.

Were any sample size calculations carried out?

Results

I think Table 1 in S4 should be a table in the main text rather than a supporting file.

Historical figures: What was the source of the data described in this section? Are these figures publically available or were they extracted from a hospital database for the study? They form the basis of a key conclusion in the study so I think it’s important to include a section in Methods to outline the data source.

Page 12: Please use the ρ coefficient to denote the results of Spearman’s rho instead of “calculated correlation co-efficient”

Discussion

Page 13, line 629: “appear outweighed” This phrasing is difficult to understand.

One element that I feel is missing from the research is information on the socioeconomic profile of the parents (education, job status, income etc.) as this is well known to influence health seeking. I think this is an important limitation and needs to be recognised or fleshed out in the discussion. Maybe information about the location of or population the hospital serves?

This research took place in November 2020 when the pandemic had been ongoing for 9 months and people had had time to adjust to living in a pandemic, restrictions, mask wearing etc. Is it possible that this may have had an impact on the results?

Are there any recommendations for policy or planning arising from the research? This would be particularly valuable with regards to future planning for public health emergencies.

Supplementary file 4:

Table 1: It’s not clear why you would expect there to be a difference between those who completed the questionnaire only, interview only or both. I think this should be collapsed into one column and this table brought into the main text.

Table 2: As mentioned previously, the source of this historical data needs to be included in the methods

Table 3: As with table 1, it’s not clear why you would expect there to be a difference between the two groups. As a result, I don’t think Table 3 is needed at all and would recommend removing.

**Reviewer #2:** Thank you so much for the opportunity to review the paper Caregiver perceptions and experiences of Paediatric Emergency Department attendance during the COVID-19 Pandemic: a mixed-methods study. This qualitative paper covers an important topic in public health literature, as it focuses in decision-making of parents to attend paediatric emergency care during the current COVID-19 pandemic. I applaud the authors for conducting this research during this challenging time and I think the results are important and relevant. Unfortunately, for this paper to be accepted for publication, several major changes need to be made, which I will outline per section below.

1. To start, a general comment of editing. This paper needs to be proofread, as there are several grammatical errors in the paper. Please attend to this.

2. The abstract reads well, but the conclusion is vague (also in the paper). Please clarify what the take-home message is of the paper and what we can learn from the your findings. What should policy-makers, health workers or parents do differently?

3. The introduction is well-written, but some of the studies mentioned in the background do not show a significant delay in seeking care for children. This does not reinforce your objectives of doing this study. Is there anything else in the examples you can highlight that are of interest to your study? This links to my concerns about the overall objectives of the study, as the background does not stress the concerns at length, and therefore, you need to reinforce your argument with using other examples. For instance, what were the qualitative outcomes in reference 26? And how are these different or similar to your work?

4. The methods are well-described, but I was wondering why data collection was only conducted during office hours and during the week? Maybe parents coming in the weekend or evening might have delayed attending the emergency room for different reasons or waited longer? I am also curious to read more about the questions asked in the interview, you only mention information provision as a topic, which seems limited. I would also like to read which emerging themes derived during the data analysis process. I would also add the specifics of the participants and the length of the interviews to the methods, instead of the findings. Also, the interviews were really short, was this time sufficient for the researchers? And why, or why not?

5. The findings are relevant, but I wonder if you can shorten theme 1. This speaks about the general experiences of COVID-19 and has some interesting quotes, but they are not related to the main topic. I would emphasize the findings of Theme 2 more, as they link well to your research questions and are well described. The theme also gives a gives a good impression of the confusion and fear of parents and their decision-making process of seeing a health care provider for their children and trying to keep other family members safe. We can learn a lot about that. In Theme 3, I was expecting to read more about parents’ interactions with health workers, such as being judged for coming in or feeling inadequate as parents, but they are not described in the theme. Only the waiting times were discussed in the theme, which seems limited.

6. The discussion of the findings is limited, which might be attributed to the novelty of the study, but I would have liked to see a comparison of your findings with studies conducted in other countries. There is literature around delayed decision-making for getting immunisations for children for instance, which has been studied in several countries. I also have several remaining questions that seem relevant, including; how were other paediatric departments in government clinics attended in other countries, or parts of the UK? And what can we learn from your findings? If we get into a similar situation, like COVID-19, how can paediatric communication be improved? Or what should be teach young parents about attending care during a national crisis. How can you convince parents that they are not ‘overburdening the NHS’ and reassure them that their own health and that of their children is most important?

Overall, I think this study is relevant and results should be published, but several improvements need to me made to strengthen the aim, objectives and outcomes of the paper.

**Reviewer #3: T**he authors have written on a relevant topic.

The following are my few comments:

Abstract

Methods: More information should be provided here, eg: method of selection of participants (sampling technique), method of data analysis/ analysis carried out

Result: Quantitative result appears too scanty here, add a little more details eg demographic characteristics.

Main Body

Methods: Line 127, Kindly provide the name of the study design used in this study, perhaps read up types of study design and identify where this falls under. What was written is more of a description of what was done.

Results of quantitative survey should also be presented in tables.

---

## [Author Response · Author response to Decision Letter 0]

23 May 2022

We have included responses below, but these can be viewed more easily in two files attached (cover letter: responses to both editor and reviewer comments; responses to reviewers: responses to reviewers only). 

Editorial requests, formatting amendments and authors comments and revisions

Comments Responses/Revisions 

https://journals.plos.org/plosone/s/fileid=ba62/PLOSOne_formatting_sample_title_authors_affiliations.pdf".

We confirm the manuscript meets PLOS ONE’s style requirements.

2. We noted in your submission details that a portion of your manuscript may have been presented or published elsewhere. [Initial findings of quantitative and qualitative work presented at the RCPCH online conference 2021 and published abstracts in Archives of Disease in Childhood. Both have been uploaded as "Related Manuscript" file types. These are not dual publications as these were top-line initial findings presented for each element of the study; here, our paper presents a complete overview of both the quantitative and qualitative studies.] Please clarify whether this [conference proceeding or publication] was peer-reviewed and formally published. If this work was previously peer-reviewed and published, in the cover letter please provide the reason that this work does not constitute dual publication and should be included in the current manuscript This work does not constitute dual publication as the abstract that we referred to was not peer-reviewed. This submission is the first time our complete results are being presented for peer-reviewed publication. 

In addition to including as “other/ related manuscript types” they can be found online here: 1683 Parental perceptions of paediatric emergency departmental attendance in children during the COVID-19 pandemic in UK (PPEDiC). The qualitative view | Archives of Disease in Childhood (bmj.com)

1639 Parental perceptions of paediatric emergency departmental attendance in children during the COVID-19 pandemic in UK (PPEDiC). The quantitative outcomes | Archives of Disease in Childhood (bmj.com)

We will update your Data Availability statement to reflect the information you provide in your cover letter. The participants of this study did not agree for their data to be shared publicly, so due to ethical concerns, we have not made the data publicly available. 

The data may be made available to interested parties who may contact the NHS Cambridge and Hertfordshire Review Committee at cambsandherts.rec@hra.nhs.uk who may approve any requests for access to date. 

Additional Editor Comments

Comments Responses/Revisions (line numbers according to clean copy)

 1. Abstract: please remove the "discussion" section from the Abstract Discussion section has been moved to the conclusion section of Abstract.

2.This is a mixed method study design, however, it reads like a multiple method study as there is no integration of the quantitative and the qualitative study findings (quantitative and qualitative results were analysed and reported separately in this manuscript). Integration is the key difference between mixed-method and multiple method study design, so please carefully address this issue in your revised manuscript. We have addressed this comment by integrating the results to allow them to speak with each other and provide a stronger cohesive voice in the resubmission. Please see Mixed Methods Results section (pages 29-31, lines 546-616).

3. Have the authors made all data underlying the findings in their manuscript fully available?

The PLOS Data policy requires authors to make all data underlying the findings described in their manuscript fully available without restriction, with rare exception (please refer to the Data Availability Statement in the manuscript PDF file). The data should be provided as part of the manuscript or its supporting information, or deposited to a public repository. For example, in addition to summary statistics, the data points behind means, medians and variance measures should be available. If there are restrictions on publicly sharing data—e.g. participant privacy or use of data from a third party—those must be specified. Anonymised transcripts are stored in an encrypted and password-protected environment only accessible to the research team (in line with ethics agreement). Data relating to quantitative findings are reported in the paper and supplementary files. No additional data available.

While revising your submission, please upload your figure files to the Preflight Analysis and Conversion Engine (PACE) digital diagnostic tool, https://pacev2.apexcovantage.com/. PACE helps ensure that figures meet PLOS requirements. To use PACE, you must first register as a user. Registration is free. Then, login and navigate to the UPLOAD tab, where you will find detailed instructions on how to use the tool. If you encounter any issues or have any questions when using PACE, please email PLOS at figures@plos.org. Please note that Supporting Information files do not need this step. We have uploaded to PACE, reviewed and downloaded an acceptable TIF file (resolution has been changed to 300 PPI).

Reviewer comments and author responses and revisions

Reviewer 1 comment Author’s response and revisions (line numbers according to clean copy)

Introduction

When identifying gaps in the literature, the authors state “as the pandemic continues there is a need to understand whether the same patterns of behaviour are being seen in emergency care, particularly whether young children are experiencing unintended consequences due to delayed presentation”. However, it is unclear how the present design and results identifies unintended consequences so I would suggest re-wording this as it may be misleading for the reader. This has been revised to acknowledge that our study is not designed to identify unintended consequences and has been removed for clarity (see lines 107-114).

It is designed to show or demonstrate the gap in the literature around if the issues previously identified are still in existence. 

Methods

Study Design and study population:

The authors state that the mixed methods approach was selected because it “allowed for corroboration, triangulation, exploration of complementary and diverging points as well as expansion of ideas”. However, it is not clear from the results how this was carried out as it appears that the quantitative and qualitative data were analysed separately. It could be that I’m interpreting this incorrectly but in that case, this section needs to be re-worded. An additional segment has been added to the results specifically integrating the quantitative and qualitative results in line with a parallel convergent mixed method approach where integration of the data occurs in the analysis stage (see lines 546-616).

Were all attendees to the ED during the timeframe described asked to participate?

If not, how were they selected and was there a risk of any bias? 

What was the response rate? The recruitment process has been made clearer. It now details how the study recruitment evolved from a planned probability systematic sampling to a non-probability convenience sampling method in the methods section (see lines 142-166).

The limitations of this change in recruitment have been discussed in the discussion section specifically about the risk of introduction of bias (see lines 697-703).

More information of the design of the survey is needed. Was it piloted ahead of the study? 

Any information on validity? The design of the survey looking at questions asked was explored later in the methods, but this wording has now been moved to the design section. 

The survey was designed by the research team and not piloted beforehand; this is discussed in the limitations section in more detail alongside how face validity was not achieved because of this omission. The interview guide was reviewed by an expert team and piloted beforehand. All of this is detailed in the Methods (see lines 169-186).

The authors mention that triage scores and outcomes were noted. Were these extracted from medical notes? If so, this needs to be stated as a source of data. This has been removed from the study as a separate data source as it was felt to not be adding to the overall story. 

Were any sample size calculations carried out? For the quantitative component, 150-200 survey respondents were expected based on attendance figures in March/April.

For the qualitative components, we aimed to continue interviewing until theoretical saturation was met.

This is detailed in the Methods (see lines 163-166).

I think Table 1 in S4 should be a table in the main text rather than a supporting file. This has been moved and formatted to meet journal requirements.

Historical figures: What was the source of the data described in this section? Are these figures publically available or were they extracted from a hospital database for the study? They form the basis of a key conclusion in the study so I think it’s important to include a section in Methods to outline the data source. The historical data sources were accessed from the Business Team and have been identified in the manuscript (see lines 188-190). In addition, these sources are not available to the public and this has been stated also. (see lines 188-190).

Page 12: Please use the ρ coefficient to denote the results of Spearman’s rho instead of “calculated correlation co-efficient” This has now been specified in the paper (see line 254).

Page 13, line 629: “appear outweighed” This phrasing is difficult to understand.

 This sentence has been changed to make it clearer that even if caregivers have concerns these are overridden and replaced by the dual influences of how their child’s health and advice/ reassurance from others can overcome these worries and support their attendance decision.

One element that I feel is missing from the research is information on the socioeconomic profile of the parents (education, job status, income etc.) as this is well known to influence health seeking. I think this is an important limitation and needs to be recognised or fleshed out in the discussion. Maybe information about the location of or population the hospital serves? In Methods section information about socioeconomic factors have been added in (see lines 137-139).

In discussion section a line has been added around the socioeconomic profile for future studies with further evidence provided around how different subgroups have been affected in the pandemic (see lines 729-734).

This research took place in November 2020 when the pandemic had been ongoing for 9 months and people had had time to adjust to living in a pandemic, restrictions, mask wearing etc. Is it possible that this may have had an impact on the results? We have discussed the temporal nature of the findings in the Results section, however, have added additional sentences in the Discussion section (under key findings) (see lines 628-632).

Are there any recommendations for policy or planning arising from the research? This would be particularly valuable with regards to future planning for public health emergencies. The following recommendations have been added in based on our study and also from reviewing the literature around public health messaging in emergency situations: 

- Tailoring specific information to groups / audiences 

- Engagement of community leaders 

- Rectifying any miscommunications early 

- Medical professionals being a trusted source

Table 1: It’s not clear why you would expect there to be a difference between those who completed the questionnaire only, interview only or both. I think this should be collapsed into one column and this table brought into the main text.

 The purpose of the table was to delineate that there was no difference. However, on reflection we have realised this may be confusing for readers and given we have the separate table of caregivers who participated in the interview the reader should be able to infer there is no difference between the two groups and so we have acted on your suggestion. 

Table 2: As mentioned previously, the source of this historical data needs to be included in the methods The source of the historical data has been identified as contacting the Business Analysis Team and we have outlined this is not available to the public. 

Table 3: As with table 1, it’s not clear why you would expect there to be a difference between the two groups. As a result, I don’t think Table 3 is needed at all and would recommend removing. As outlined previously the idea of including the results was to show no difference between the groups. We have decided to keep the data to support the findings reported in the main manuscript but agree it should remain in the Supplementary File. 

 

Reviewer 2 comment Author’s response and revisions (line numbers according to clean copy)

1. To start, a general comment of editing. This paper needs to be proofread, as there are several grammatical errors in the paper. Please attend to this. We have proofread the paper.

2. The abstract reads well, but the conclusion is vague (also in the paper). Please clarify what the take-home message is of the paper and what we can learn from the your findings. What should policy-makers, health workers or parents do differently? We have revised the Conclusion section of the Abstract on page 3 and the Conclusion section in the main text on page 39 to better communicate our recommendations. However, the purpose of this manuscript is to report findings and it is beyond the scope of this study to dictate specific messages tailored to different groups to improve care seeking. The suggestions we do make are grounded in the data- for instance, that caregivers should not delay care seeking, emphasising the safety of hospital spaces in terms of SARS-COV-2 transmission.

3. The introduction is well-written, but some of the studies mentioned in the background do not show a significant delay in seeking care for children. This does not reinforce your objectives of doing this study. Is there anything else in the examples you can highlight that are of interest to your study? This links to my concerns about the overall objectives of the study, as the background does not stress the concerns at length, and therefore, you need to reinforce your argument with using other examples. For instance, what were the qualitative outcomes in reference 26? And how are these different or similar to your work?

 We have added references indicating significant declines in global settings including Germany [8], Ireland [9] and Greece [10]. We have included the studies that do not show significant delays to emphasise that the picture remains mixed and context-specific which highlights why our study is needed for this specific population.

We agree that including some more information on the qualitative outcomes from the Watson et al. 2021 paper [now reference 29] would be helpful, so we have this in the Introduction. We have reflected on our findings compared with Watson’s paper the Discussion.

4. The methods are well-described, but I was wondering why data collection was only conducted during office hours and during the week? Maybe parents coming in the weekend or evening might have delayed attending the emergency room for different reasons or waited longer? We have referenced that this was a limitation to our study in the limitations section. 

Historically research has only been conducted in hours 9-5 at Lewisham and Greenwich NHS trust in the paediatric department due to the contractual obligations of the research team. Whilst this research was conducted voluntarily, by those involved in data collection, for uniformity and in line with previous studies the recruitment occurred during ‘office-hours’. 

I am also curious to read more about the questions asked in the interview, you only mention information provision as a topic, which seems limited. 

I would also like to read which emerging themes derived during the data analysis process. 

I would also add the specifics of the participants and the length of the interviews to the methods, instead of the findings. 

Also, the interviews were really short, was this time sufficient for the researchers? And why, or why not? The interview topic guide which lists the questions asked during the interviews is provided in the Supplementary Information (Item 2).

The themes which emerged from the data analysis are those which are presented as headings and described in the Results section of the manuscript, also, an overview of these are depicted in Figure 1. 

Thank you for the suggestion of moving this table to the methods. After some deliberation we decided to keep the participant characteristics and nature of interviews (including length) in the Results section to be consistent with the presentation with the quantitative findings. We feel that this best supports the narrative and flow of the paper.

The interviews ranged from 10-45 minutes. As the interviews were conducted in a busy Paediatric A&E department whilst participants’ children were waiting or in between receiving emergency care, the length of time that participants were willing to participate in an interview varied and may have been a shorter length than if they were interviewed at a less urgent time. Despite this, as the richness and range of the results illustrate, valuable findings emerged from the interview data. 

5. The findings are relevant, but I wonder if you can shorten theme 1. This speaks about the general experiences of COVID-19 and has some interesting quotes, but they are not related to the main topic. I would emphasize the findings of Theme 2 more, as they link well to your research questions and are well described. The theme also gives a gives a good impression of the confusion and fear of parents and their decision-making process of seeing a health care provider for their children and trying to keep other family members safe. We can learn a lot about that. 

In Theme 3, I was expecting to read more about parents’ interactions with health workers, such as being judged for coming in or feeling inadequate as parents, but they are not described in the theme. Only the waiting times were discussed in the theme, which seems limited. We agree with the comment made by Reviewer 2 and have shortened theme 1.

We agree and have emphasised theme 2 more in the Discussion.

Exploring the interactions of caregivers and healthcare workers was not within the scope of the study (nor interview topic guide) and did not emerge as a theme from the interviews.

6. The discussion of the findings is limited, which might be attributed to the novelty of the study, but I would have liked to see a comparison of your findings with studies conducted in other countries. 

There is literature around delayed decision-making for getting immunisations for children for instance, which has been studied in several countries. 

I also have several remaining questions that seem relevant, including; how were other paediatric departments in government clinics attended in other countries, or parts of the UK?

And what can we learn from your findings? 

If we get into a similar situation, like COVID-19, how can paediatric communication be improved? Or what should be teach young parents about attending care during a national crisis. How can you convince parents that they are not ‘overburdening the NHS’ and reassure them that their own health and that of their children is most important?

Overall, I think this study is relevant and results should be published, but several improvements need to me made to strengthen the aim, objectives and outcomes of the paper. We have added findings from other global studies in the Introduction section [8-10] and have added an additional reference [43] in the Discussion to unpick how caregiver characteristics may explain delayed presentations to PED.

We have added detail on the description of PED attendance in Greece in lines 69-73 and existing literature in the UK/England in lines 80-90.

We have added recommendations for risk communication in lines 729-757 as well as conclusion section.

Reviewer 3 comments Author’s response and revisions

Methods: More information should be provided here, 

eg: method of selection of participants (sampling technique), method of data analysis/ analysis carried out We have addressed and added more clarity around the sampling technique used in the study. 

Result: Quantitative result appears too scanty here, add a little more details eg demographic characteristics. We have restructured the results section and added a subheading and results to refer to demographics.

We have re-worked Table 2. We have also added additional analyses using ANOVA test to compare mean ages between caregiver ethnicity

Methods: Line 127, Kindly provide the name of the study design used in this study, perhaps read up types of study design and identify where this falls under. What was written is more of a description of what was done. Thank you for prompting us to be more explicit with the type of study design used. It is a mixed methods parallel convergent study design. This is stated in lines 117-122. 

Results of quantitative survey should also be presented in tables. Please see Supplementary File 4.

---

## [Decision Letter · Decision Letter 1]

22 Jul 2022

PONE-D-21-33341R1Caregiver perceptions and experiences of paediatric emergency department attendance during the COVID-19 pandemic: a mixed-methods studyPLOS ONE

Dear Dr. -Nyang'wa,

Thank you for submitting your manuscript to PLOS ONE. After careful consideration, we feel that it has merit but does not fully meet PLOS ONE’s publication criteria as it currently stands. Therefore, we invite you to submit a revised version of the manuscript that addresses the points raised during the review process.

We look forward to receiving your revised manuscript.

Kind regards,

Alison Wang

Academic Editor

PLOS ONE

Journal Requirements:

Reviewers' comments:

Reviewer's Responses to Questions

**Comments to the Author**

1. If the authors have adequately addressed your comments raised in a previous round of review and you feel that this manuscript is now acceptable for publication, you may indicate that here to bypass the “Comments to the Author” section, enter your conflict of interest statement in the “Confidential to Editor” section, and submit your "Accept" recommendation.

Reviewer #3: All comments have been addressed

Reviewer #4: All comments have been addressed

2. Is the manuscript technically sound, and do the data support the conclusions?

Reviewer #3: Yes

Reviewer #4: Yes

3. Has the statistical analysis been performed appropriately and rigorously? 

Reviewer #3: Yes

Reviewer #4: Yes

4. Have the authors made all data underlying the findings in their manuscript fully available?

Reviewer #3: Yes

Reviewer #4: Yes

5. Is the manuscript presented in an intelligible fashion and written in standard English?

Reviewer #3: Yes

Reviewer #4: Yes

6. Review Comments to the Author

Reviewer #3: Overall comment:

I think this paper has drastically improved and I am happy with the changes made in the updated version. I am happy for this paper to be accepted, although I have made some additional suggestions below. Once these are considered, I think this paper will make an excellent contribution to body of knowledge focusing on the impact of the COVID-19 pandemic in the global North.

Comments:

1. Abstract: in the methods, you speak about data triangulation, but in the mixed methods study, you only used 2 different methods; survey and interviews. How was the data triangulated? And more importantly, how was the data analysed and synthesized?

2. Line 67: use a different word to start the sentence. Despite this – does not link well to the previous paragraph, unless you say: ‘despite hospitals remaining open throughout the various lockdowns, the was a decreased …..’

3. Line 102: ‘was driven by fear and knowledge’ – can you be more specific here? What kind of knowledge are we speaking about…. Or a lack of knowledge maybe? This seems relevant to the study.

4. Line 161: consider rewriting: ‘that they were too worried about and wanted to focus on their child.’ Just change into; feeling too worried about their child.

5. Line 188: I understand that the data cannot be disclosed, but how was the ‘historical data’ used? To see if the child had been attending the ER before COVID, or to understand illness and symptoms?

6. Line 205: what themes derived out of the analysis? And how was reflexivity and positionality of the project staff being explored and negotiated within the team? I would like to read more about this, as it is an important aspect of data analysis.

7. Line 298: I think it would be great to have a quote that illustrates the ‘new normal’.

8. Line 309: Similar to comment above. A quote from ID12 or ID20 speaking about the differences in illness symptoms between flu and COVID could strengthen this paper

9. Line 350: A quote about the confusion of the rules would be a benefit to this paper

10. Line 376: A last sentence can be added here that explains that the interviewees had been living with the pandemic for 6 months at the time of the interview, so had a chance to adapt to the ‘new normal’.

11. Line 395: I would like to read a few quotes here too, especially about the ‘parents’ instinct or ‘gut feeling’.

12. Line 739 – 741: this sentence does not flow well, and I am not sure what the authors are trying to say. Please rewrite/simplify

Reviewer #4: (No Response)

7. PLOS authors have the option to publish the peer review history of their article (what does this mean?). If published, this will include your full peer review and any attached files.

Reviewer #3: No

Reviewer #4: No

---

## [Author Response · Author response to Decision Letter 1]

3 Sep 2022

We have submitted a document outlining the changes made in response to Reviewer 3 comments titled 'response to reviewers'. We have tried to meet all the points that were asked.

---

## [Editor Report · Decision Letter 2]

29 Sep 2022

Caregiver perceptions and experiences of paediatric emergency department attendance during the COVID-19 pandemic: a mixed-methods study

PONE-D-21-33341R2

Dear Dr. Maggie Nyirenda,

We’re pleased to inform you that your manuscript has been judged scientifically suitable for publication and will be formally accepted for publication once it meets all outstanding technical requirements.

Kind regards,

Alison Wang

Academic Editor

PLOS ONE
---

## [Editor Report · Acceptance letter]

8 Nov 2022

PONE-D-21-33341R2 

Caregiver perceptions and experiences of paediatric emergency department attendance during the COVID-19 pandemic: a mixed-methods study 

Dear Dr. -Nyang'wa:

I'm pleased to inform you that your manuscript has been deemed suitable for publication in PLOS ONE. Congratulations! Your manuscript is now with our production department. 

Kind regards, 

on behalf of

Dr. Tao (Alison) Wang 

Academic Editor

PLOS ONE